# Three dimensional analysis of hip joint reaction force using Q Hip Force (AQHF) software: Implication as a diagnostic tool

**Amany Eid Abd El-Tawab**[1,2], **Aisha Farhana**[3]*

**1** Department of Physical Therapy and Health Rehabilitation, College of Applied Medical Science, Jouf University, Aljouf, Saudi Arabia, **2** Faculty of Physical Therapy, Biomechanics Department, Cairo University, Cairo, Egypt, **3** Department of Clinical Laboratory Sciences, College of Applied Medical Sciences, Jouf University, Aljouf, Saudi Arabia

* aishafarhana512@gmail.com

**Data Availability Statement:** All relevant data are within the paper.

**Funding:** The author(s) received no specific funding for this work.

## Abstract

Assessment of hip joint reaction force (JRF) is one of the analytical methods that can enable an understanding of the healthy walking index and the propensity towards disease. In this study, we have designed software, Analysis Q Hip Force (AQHF), to analyze the data retrieved from the mathematical equations for calculating the JRF and ground reaction force (GRF) that act on the hip joint during the early part of the stance phase. The stance phase is considered the least stable sub-phase during walking on level ground, and the gait stability is sequentially minimized during walking on elevated ramps. We have calculated the JRF and GRF values of walking stances on varied inclinations. The data obtained from these calculations during walking on elevated ramps were exported from mathematical equations to Q Hip Force software as two separate values, namely the JRF data and GRF data of the hip joint. The Q Hip Force software stores the two reaction force data in a text file, which allows the import and easy readability of the analyzed data with the AQHF application. The input and output data from the AQHF software were used to investigate the effect of different walking ramps on the magnitude of the hip JRF and GRF. The result of this study demonstrates a significant correlation between the JRF/GRF values and healthy walking indices till a ramp elevation of 70˚. The software is designed to calculate and extrapolate data to analyze the possibility of stress in the hip joint. The framework developed in this study shows promise for preclinical and clinical applications. Studies are underway to use the results of JRF and GRF values as a diagnostic and prognostic tools in different diseases.

## 1. Introduction

Many three-dimensional (3D) joint, limb, and musculoskeletal models have been developed for gait analysis and tested in healthy and disabled conditions and in individuals with prostheses [1–5]. Such studies help us understand the motor patterns categorized as satisfactory or impaired gait conditions. Impaired gait pattern is one of the primary symptoms associated with a plethora of diseases such as cervical spondylotic myelopathy, arthritis, multiple sclerosis,

**Competing interests:** The authors have declared that no competing interests exist.

Meniere's disease. In some diseases, a detailed assessment is usually carried out using the three-dimensional gait analysis (3DGA) tools. However, the administration and reliability of quite a few of these tools have not been established and remain uncommon in clinical settings. One of the reasons is the use of costly and cumbersome devices for procuring the data besides being computationally expensive.

Upslope walking requires raising the knee to a higher position compared to level walking, which is facilitated by a well-coordinated neuromuscular system [6]. The center of mass (COM) has to be adjusted to overcome the gravitational force and raised further upwards to allow forward movement. Gait assessments during walking up the slopes potentially help in understanding many associated derangements of the joints, which may not be clearly identifiable during leveled walking. Additionally, upslope walking assessments are also useful in optimizing mobility in various pathological conditions such as spinal cord disabilities [7]. In the present study, we investigated the gait parameters of healthy subjects during walking up ramps of inclinations between 0˚-15˚. Wherein, the 0˚ represents level ground walking reflecting the absence of inclined surface during walking. We used the motion analysis system that uses marker-based motion planes and a force platform to record the kinematic and kinetic gait parameters in a subject-specific investigation method. The system consisted of six high-velocity infrared Pro-Reflex cameras and an AMTI (Advanced Mechanical Technology Inc., USA) force plate added as an impediment in the center of a walkway. It has a width is 40 cm and a length is 60 cm. The motion frequency rate is 120 Hz. The results were analyzed using the Q-gait analysis software. We observed a significant variation in the vertical COM displacement between subjects during the single-limb support phase while walking up different ramps. An increase in COM during level walking ascended on an average of 4.4 cm during the first half of the stance phase and descended on an average of 4.4 cm during the second half of the stance phase. According to previous reports, the highest point of this vertical displacement occurred at about 30% and 60% of the gait cycle [8,9]. Furthermore, the angular displacement of the hip joint in the frontal plane during the walking phase is the most variable and is most affected during walking on uneven terrain and inclinations [10]. Hence an overall assessment of gait while walking upslope requires calculating the magnitude of GRF and JRF in three dimensions. The resultant effect on the hip joint is thus assessed during single limb support and entered into the complex equations. In this study, we designed the Analysis Q hip force (AQHF) software and used the gait data as input for each subject to extract the final results according to body weight of the subjects and the angle of the inclined ramp. This framework assimilates the ease of data acquisition and computational ability, thus holding better prospects for use in clinical settings for the assessment of Hip forces in a subject- specific manner.

## 2. Results

This study was conducted on twenty male and twenty female university students to investigate the effect of walking ramps (0˚- 5˚- 10˚- 15˚) on hip JRF and the GRF. Mixed Design MANOVA was used with the resultant hip JRF as the first dependent factor, the resultant GRF as the second dependent factor. The independent factors were the degrees of the walking ramps and gender. The least significant difference (LSD) multiple comparison post hoc test was used to determine the significant difference between the mean values of the dependent factors during walking up ramps and compare the changes that occur in these factors between male and female subjects. All statistical analyses were conducted using SPSS for Windows, version 17.0.0 (SPSS, Inc., Chicago, IL). The alpha level of significance was set at 0.05. Effect of walking ramps (Factor A) on 3D hip JRF was calculated using the AQHF application. The result obtained from the analysis showed that the resultant mean value of the hip JRF (as a multiple

of body weight) in 3D during single-limb support on ramps of (0˚, 5˚, 10˚, and 15˚) was 119.4 (±16.1) body weight (BW), 129.5 (±17.4) BW, 138.6 (±18.2) BW and 149 (±19.6) BW respectively. Using the factorial design (Mixed Design MANOVA) test, a significant difference was found among the four tested ramps for the mean value of the resultant hip JRF ($p < 0.05$).

Meanwhile, paired comparison using the least significant difference (LSD) multiple comparison post hoc test revealed that there was a considerable difference between every two ramps except for the difference between 0˚,5˚, and 5˚,10˚, which revealed no statistically significant difference. Table 1 and figure (Fig 1) summarize these results.

## 2.1 Effect of gender (Factor B) on 3D hip JRF obtained from AQHF Software

Table 2 and figure (Fig 2) summarize the mean values of the resultant 3D hip JRF (BW) in both males and females while walking over level ground and inclinations of 5˚, 10˚, and 15˚. Results indicated that 3D hip JRF in males is higher than in females in all walking inclinations.

Mixed Design MANOVA revealed significant differences between males and females among all tested inclinations ($p < 0.05$). Statistically, a significant difference was observed in males between every two ramps except between 0˚ and ramp 5˚ and ramp 5˚ and ramp 10˚. In addition, post hoc analysis showed that there was only a significant difference in females between 0˚ and ramp 10˚ and between 0˚ and ramp 15˚.

## 2.2 The interaction effect of ramps and gender on the resultant hip JRF

Using the factorial designs (Mixed Design MANOVA test), a significant interaction ($p < 0.05$) was found between the ramp and the gender on the mean value of the resultant hip JRF (Table 2). This was supported by the Posthoc analysis (Table 3).

Moreover, paired comparison using the LSD post hoc test revealed that there were significant differences between males (M) and females (F) and between each ramp and other in the mean value of the resultant hip JRF (Fig 3).

Female

## 3. Discussion

The JRF results obtained in our study showed a significant difference among the tested ramps for the resultant of the right hip JRF during the single support phase that represents from 30%

**Table 1. Descriptive statistics and Mixed Design MANOVA for 3D hip JRF during single-limb support on the four tested ramps (0˚, 5˚, 10˚, and 15˚) as a multiple of body weight (BW) for all the tested subjects.**

| 3D hip joint reaction force (BW) | | | |
|---|---|---|---|
| Ramp | **0˚** | **5˚** | **10˚** | **15˚** |
| X± SD | 119.4(±16.1) | 129.5 (±17.4) | 138.6 (±18.2) | 149 (±19.6) |
| **Mixed Design MANOVA** | | | |
| F = 13.85 | | $p = 0.000^*$ | |
| **LSD multiple comparison test** | | | |
| 0˚ vs 5˚ | | $p > 0.05$ | |
| 0˚ vs 10˚ | | $p < 0.05^*$ | |
| 0˚ vs 15˚ | | $p < 0.05^*$ | |
| 5˚ vs 10˚ | | $p > 0.05$ | |
| 5˚ vs 15˚ | | $p < 0.05^*$ | |
| 10˚ vs 15˚ | | $P < 0.05^*$ | |

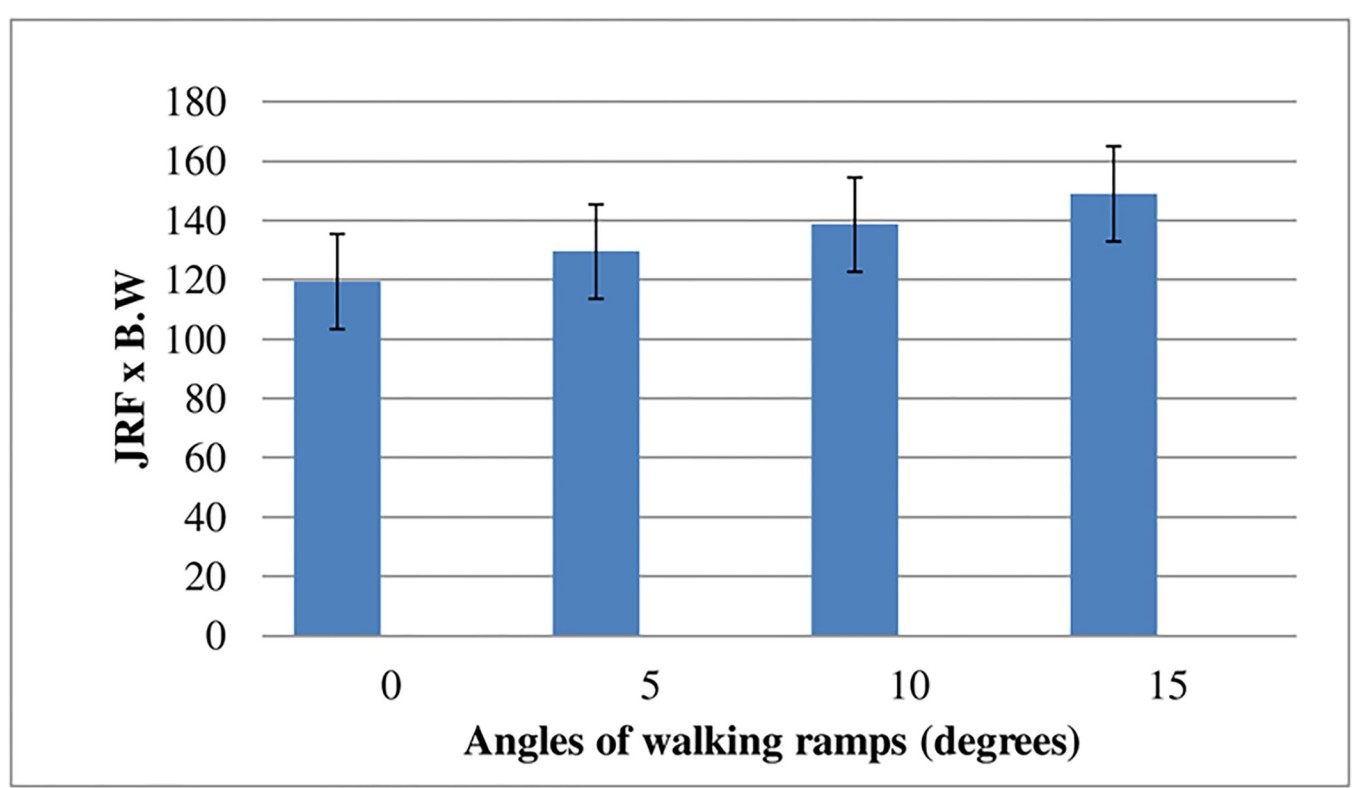

**Fig 1. The variation of the hip joint reaction force during single-limb support on the four walking ramps.**

up to 60% of the whole gait cycle, which starts with initial contact of the foot and ends at terminal swing phase of the same foot. (Table 1, Fig 1). In addition, paired comparison of LSD revealed significant differences among walking up the ramps (Table 1). According to the present study, the vertical component (Jz) of the JRF has the highest magnitude compared to anteroposterior and mediolateral components. Our results are supported by studies from other groups who assessed the hip and knee joint kinetics to estimate bone-on-bone contact forces during level walking and stair climbing in healthy subjects with a subject-specific joint model approach [11,12]. The components of hip forces were Fz, Fx, and Fy in the represent the vertical, mediolateral, and anteroposterior directions. Their results revealed a significant difference in the hip joint contact forces during stair ascent compared to its value during level walking comparable to our results. The mean value of the (posterior-anterior) force at the hip joint during stair ascent was estimated to be twice its value during level walking. Also, the magnitude of the vertical hip JRF (Fz) achieved the most significant value compared to the mediolateral (Fx) and anteroposterior (Fy) reactive force during single-limb support. Furthermore, Salam M. Elhafez (2019), also demonstrated that the joints of the lower extremities were most loaded during the single supporting phase of the gait cycle [13]. They observed the reaction forces of these joints to be significantly increased at this sub-phase while walking. The JRF acting at the hip joint can be correlated to motion and stability for prosthetic impingement if the acetabular abduction angles are altered from the normal 45–50 degrees [14,15]. Acetabular cups with an angle of > 50˚ demonstrated a volumetric wear rate of 160 mm 3 per year. This exceeds the critical rate of normal wear and tear and thus likely to result in osteolysis and fixation problems due to increased JRF [16,17]. Poor positioning of the inclination angle of the acetabular cup > 50˚ might result in edge-loading and a greater rate of wear and tear

**Table 2. The table demonstrates the (a) descriptive statistics for the interaction between ramps and gender on the mean values of the resultant hip JRF while walking up ramps as a multiple of body weight (BW), (b) Mixed Design MANOVA for the resultant hip joint reaction force of males and females during single-limb support on the four walking ramps (0˚, 5˚, 10˚, 15˚).** Calculation of least significant difference (LSD) in (c) males and in (d) females.

**a. Hip joint reaction force (BW)**

| Ramp | 0˚ | | 5˚ | | 10˚ | | 15˚ | |
|---|---|---|---|---|---|---|---|---|
| Gender | Male | Female | Male | Female | Male | Female | Male | Female |
| X± SD | 121.4 (±15.7) | 115.8 (±16.9) | 131.6 (±17.42) | 127.3 (±19.5) | 138.2 (±14.6) | 136.1 (±18.9) | 155.6 (±17.8) | 137.2 (±16.4) |

**b. Mixed Design ANOVA**

| F = 8.18 | p = 0.000* |
|---|---|

**c. LSD multiple comparison test in males**

| 0˚ vs 5˚ | p > 0.05 |
|---|---|
| 0˚ vs 10˚ | p < 0.05* |
| 0˚ vs 15˚ | p < 0.05* |
| 5˚ vs 10˚ | p > 0.05 |
| 5˚ vs 15˚ | p < 0.05* |
| 10˚ vs 15˚ | p < 0.05* |

**d. LSD multiple comparison test in females**

| 0˚ vs 5˚ | p > 0.05 |
|---|---|
| 0˚ vs 10˚ | p < 0.05* |
| 0˚ vs 15˚ | p < 0.05* |
| 5˚ vs 10˚ | p > 0.05 |
| 5˚ vs 15˚ | p > 0.05 |
| 10˚ vs 15˚ | p > 0.05 |

associated with adverse biological reactions associated with metal ion release and an increase in the liability of its dislocation [18].

According to Newton's third law, the body's normal reaction force (N) acts vertically on an inclined plane. Thus, the GRF vector will always be oriented perpendicular to the inclined plane. The trunk and the pelvis remain aligned with the earth's vertical axis at all surface inclinations during single limb quiet standing [19]. Hence, the center of gravity (COG) remains constant with changing surface inclination [19]. This points that the mathematical calculation of GRF is carried out by its perpendicular orientation with an inclined plane and the alignment of its moment arm while walking on inclined surfaces during single limb standing. Hence, we have selected the sub-phase position to calculate the GRF magnitude at the hip joint using the AQHF software (Fig 1).

## 3.1 Effect of gender (factor B) on 3D hip JRF obtained from AQHF software

An increase in the mean value of the resultant hip JRF of males was observed as compared to females (Tables 2 and 3). This may be attributed to difference in the gait-related anatomy and habits between males and females [20]. Previous studies have demonstrated a significant gender difference in the gait pattern while walking. Attributed to the specialized character of a wider pelvis in females compared to males, significant difference in the gait pattern are noted [20]. Furthermore, the females walked with the pelvis tilted more anteriorly and with more up and down oblique motion. The hip joints are more flexed, adducted and internally rotated, while the knee joints have more valgus angles.

Females are shorter, both in height and leg length and they walk slower than males due to shorter stride length and narrower step width. Hence, during the stance phase, a shorter

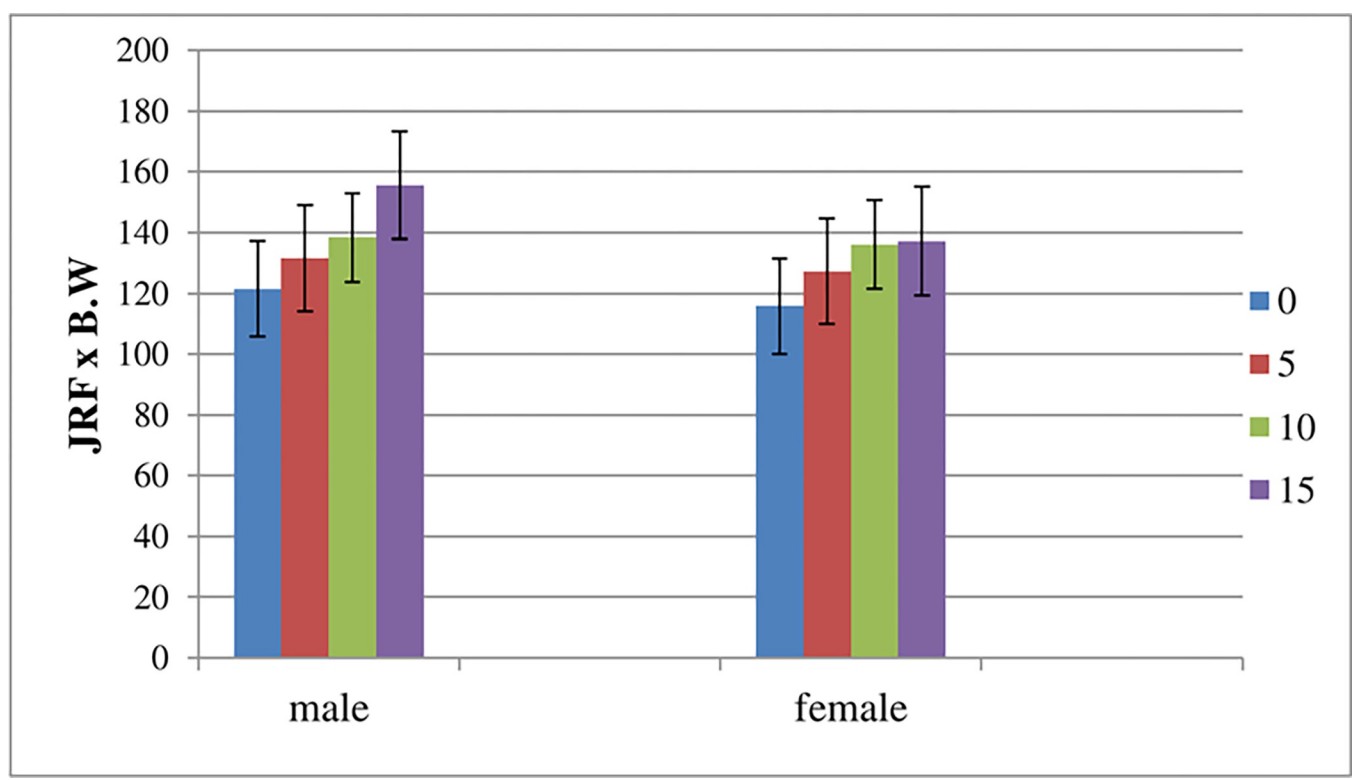

**Fig 2. The variation of the resultant hip joint reaction force between males and females during single-limb support on the four walking ramps.**

walking stride results in reduced joint compression forces acting on the lower extremities. [21]. A significant difference of hip JRF also occurs when the walking speed is reduced by 15%. This decrease in the hip JRF resulted from the reduction in the posterior joint force at the hip when the subjects walked at a speed 15% below average [21].

Further, the electromyographic JRF examination of men resulted in a large peak of seven times body weight just before toe-off when the abductors contracted to stabilize the pelvis. [22].

### 3.2 The interaction effect of (ramps and gender) on the resultant hip JRF

We found that the males exhibit a higher mean value of the resultant hip JRF than females at an elevation of 0°, 5°, 10° and 15° during single limb support and a significant effect was found on the mean value of the resultant of the hip JRF. A significant gender difference in gait patterns is demonstrated to occur while walking [23]. Comparable results were obtained by another study establishing that the the step length of females decreased compared to the step

**Table 3. The table demonstrates the post hoc test carried out on male and female subject's categories.** The test is based on the hip joint reaction force calculated for different ramp elevations as in Table 2.

**a. Post hoc test (JRF)**

| | | Males | | | |
|---|---|---|---|---|---|
| Females | Ramp | 0˚ | 5˚ | 10˚ | 15˚ |
| | 0˚ | | | | |
| | 5˚ | $p < 0.05^*$ | | | |
| | 10˚ | $p < 0.05^*$ | $p < 0.05^*$ | | |
| | 15˚ | $p < 0.05^*$ | $p < 0.05^*$ | $p < 0.05^*$ | |

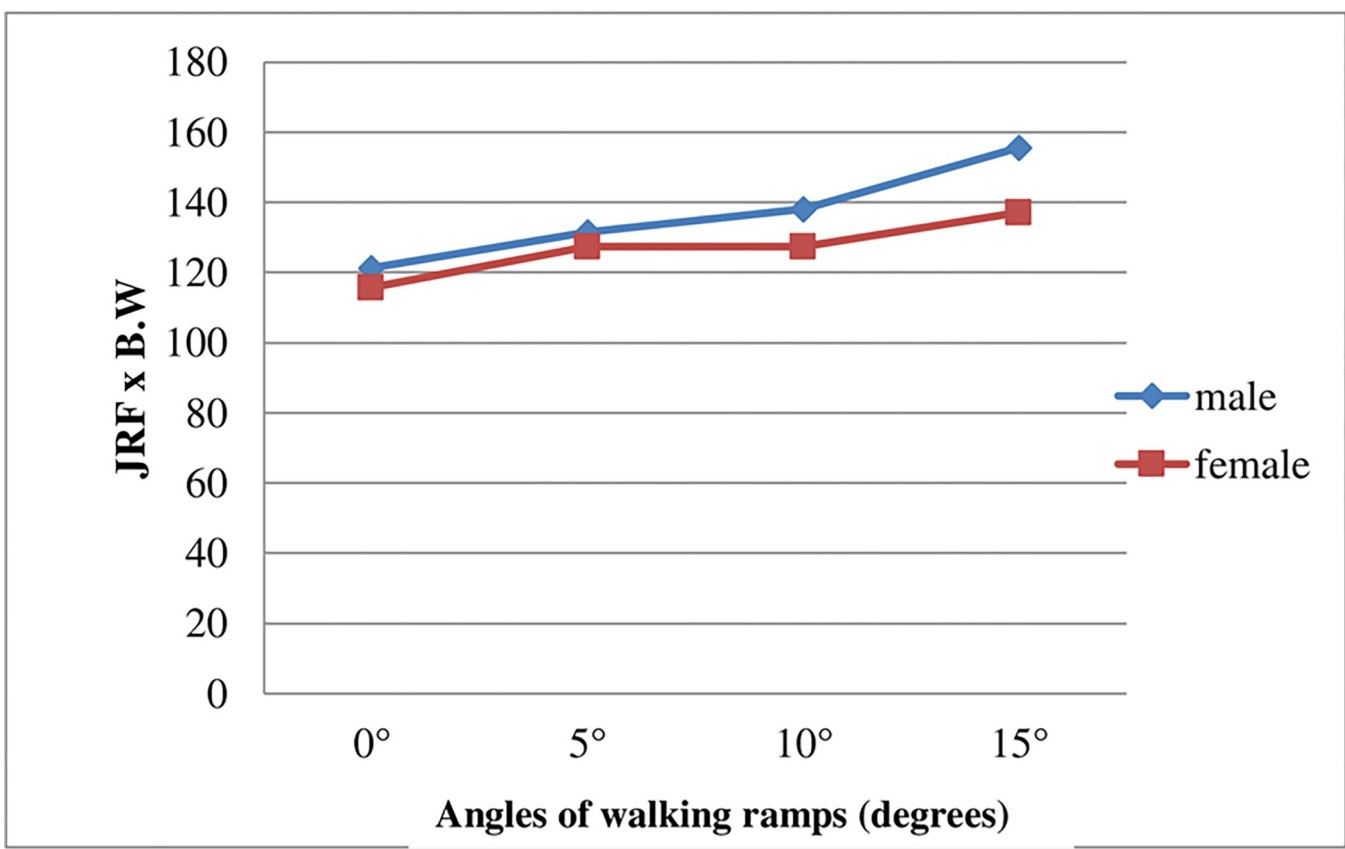

**Fig 3. The interaction in the mean difference of the right hip joint reaction force between males and females among the four walking ramps during single-limb support.**

length of males during ramp walking. This produced a reduction in the friction demand of females hence minimizing the joint compression forces of their lower extremities during the stance phase of walking [24]. Also, a short stride length probably counteracts the higher friction demand that would otherwise be needed at heel strike in upslope walking [25].

The walking speed and cadence also affect the lower extremity JRF. A differences in the JRF of the lower extremities was observed when speed and cadence were manipulated [26]. The most significant differences occurred when the subjects reduced their walking speed by 15%. Therefore, a significant interaction between walking speed and gender on hip JRF was effective while walking up the ramps (Fig 3).

Our results also indicate a negative value for the GRF output. This negative sign obtained by software shows the value of an angle, which depends on the quadrant that harbors the terminal side of θ. If θ is a second quadrant angle. Therefore, $\cos \theta = \frac{-a}{r} = -\cos \phi$ (Fig 4A and 4B) is in accordance with the previous mathematical theory proved by Lia and Won (2008). As shown in Fig 4(B) the angle between the vector (W) representing the pathway of the GRF vector "CE" with the ground in Fig 4(A) lies in the second quadrant during the single-limb support phase on different inclinations. Thus, "Wx = —W (cos θ)" as the component of the GRF in the X-direction results in a negative sign. The results of our study can also be biomechanically interpreted for hip dislocation and active forces on the hip joint.

The negative sign of the X component observed for the GRF may result from the variation in the angular displacement of the hip joint while walking up ramps. It was known that the

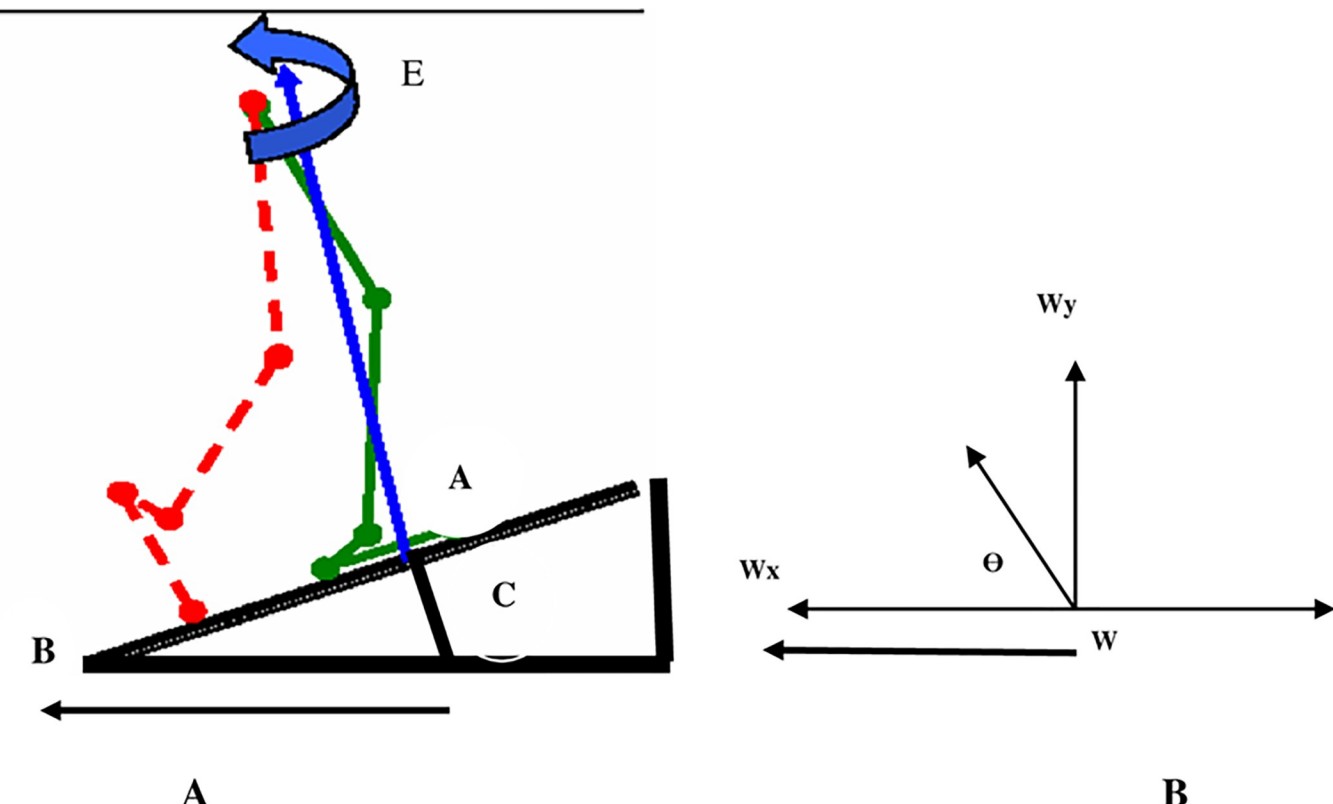

**Fig 4.** (A) The pathway of the GRF vector during single-limb support on an inclined surface. (B) θ is a second quadrant angle that lies between the vector (W) & the ground.

range of hip flexion and extension occurs on the X-axis. Thus, if the hip flexion range occurs in the inner range and has a positive sign, the range of hip extension occurs in the outer range and will have a negative sign. As the extended position of the hip joint lies in the outer range of the angular movement with a negative sign, the X components of the GRF lie in the 2nd quadrant and have the same sign.

Subsequently, the patients who suffer from recurrent posterior hip dislocation of the artificial hip may be due to the liability of the hip joint to be dislocated posteriorly, especially during standing over the affected limb in the single-limb support phase. This causes an increase in the horizontal components of the GRF acting over the hip joint at this sub-phase

According to the equation, the direction of the GRF is "Tan $\alpha = \frac{Wy}{Wx}$". As Wx = -W(cosθ). Therefore, the α of the GRF also has a negative sign. According to the above equation, we concluded that not only do the angular displacements of the hip joint in the sagittal plane affect the hip position; but also the direction of the GRF vector in relation to the horizontal plane "α," which has an important implication in determining the best position of the artificial hip. Consequently, the incidence of recurrent posterior hip dislocation will be decreased. Hence, there exists a relation between the angular displacement of the hip joint in the sagittal plane and the direction of the GRF vector at the hip joint.

### 3.3 Effect on the hip Forces

As the X-component of the abductor muscle force equals the GRF value at the hip joint (Ax = W), the GRF will gain a negative direction with the horizontal plane (Fig 5A). In

**a**

Counter clockwise moment = clockwise moment

A (abductors) X c = gravitational force (5/6 BW) X b

**As the value of b = 2c**

$\therefore$ A x c = (5/6 BW) X 2c

A = 2 (5/6 BW)

**As the gravitational force (5/6 BW) = the ground**

**reaction force (W)**

$\therefore$ A= 2W

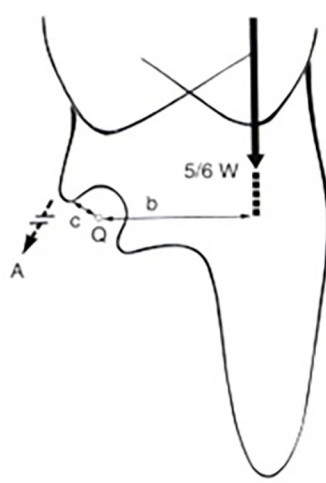

**b**

As the value of the abductors force (A) = 2W
It can be resolute into horizontal and vertical
components:

Ax= A sin 30°

Ax= 0.5 A

**As the value of A = 2W**

$\therefore$ Ax = W

Ay= A cos 30°

Ay = 0.8 A

**As the value of A = 2W**

$\therefore$ Ay= 1.7W

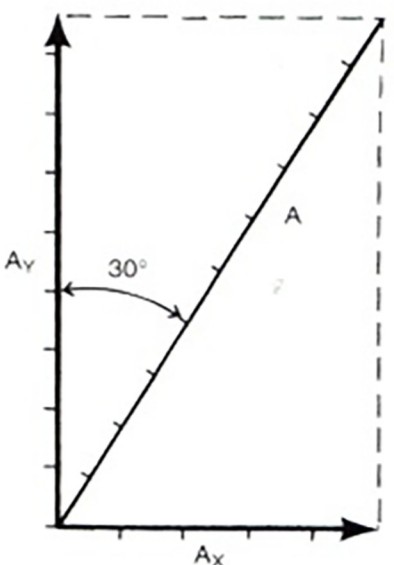

**Fig 5.** The interaction effect of ramp and gender on the hip forces (a) Calculation to derive the relationship between the gravitational force and the internal hip joint moment during single limb support. (b) The horizontal and the vertical components of the abductors force "A" (Adapted from Frankel and Nordin, 2020) [28].

addition to the value for 'A' component, which represents the abductor muscle force and equals two times the body weight, the direction of force 'A' will have a positive sign. It will also be 30° from the vertical axis (Fig 5B) [27].

### 3.4 The interaction effect (ramps and gender) on the resultant total GRF

Females exhibit a higher mean value of the resultant total ground reaction force than males at the mid stance phase of the gait cycle while walking ramps of 0°, 5°, and 15°. However, male exhibit a higher mean value than female while walking up a ramp of 10°. Statistical analysis using Mixed Design MANOVA revealed that there was a significant interaction ($p < 0.05$) between the ramp and the gender on the mean value of the resultant total ground reaction force (Table 4A). The percentage varied from 30% to 60% measured at the most critical walk

**Table 4. The tables indicate (a) the interaction between the ramps and the gender on the mean value of the resultant total ground reaction force at the mid stance phase of the gait cycle while walking up ramps.** (b) the post hoc test carried out on male and female subject's categories and is based on the hip ground reaction force calculated for different ramp elevations as in Table a.

**a. Hip ground reaction force (N)**

| Ramp | 0° | | 5° | | 10° | | 15° | |
|---|---|---|---|---|---|---|---|---|
| Gender | Male | Female | Male | Female | Male | Female | Male | Female |
| X± SD | 509.49 (±34.69) | 518.64 (±5.41) | 499.87 (±25.3) | 504.25 (±22.49) | 527.7 (±27.38) | 517.57 (±15.58) | 549.58 (±3.95) | 552.24 (±19.24) |

**Mixed Design MANOVA**

| F = 7.5 | p = 0.000* |
|---|---|

**b. Post hoc test (GRF)**

| Females | Ramp | \<Males\> 0° | 5° | 10° | 15° |
|---|---|---|---|---|---|
| | 0° | | | | |
| | 5° | P > 0.05 | | | |
| | 10° | P > 0.05 | P < 0.05* | | |
| | 15° | P < 0.05* | P < 0.05* | P < 0.05* | |

phase of the gait cycle, i.e. middle of the stance phase. Multiple comparison post hoc test revealed that there was a significant difference in the mean value of the resultant total ground reaction force between male (M) and female (F) for each of ramp of 0° and ramp of 15°, ramp of 5° and ramp of 15°, ramp of 5° and ramp of 10° and ramp of 10° and ramp of 15° at the mid stance phase of the gait cycle while walking. (Table 4B).

Biomechanically, the single-limb support phase during walking up a ramp, reaches a balancing state to maintain the pelvis level in the frontal plane and prevent its drop toward the swing extremity [29]. This is a compensatory mechanism between the abductor muscle force and the GRF at the hip joint. Thus, the negative direction of the GRF vector against the horizontal plane is compensated by the positive X-component of the abductors force to maintain a leveled pelvis [30].

However, in case of abductor muscle weakness, the sum of the abductors force in the X-direction and the gravitational forces is not equal to 0. This represents a state of imbalance wherein the gravitational force is greater than the X-component of the abductor muscle force. Finally, the sum of these forces gets directed toward the greater one [31]. The posterior orientation of the GRF vector at the hip joint in the X-axis exposes the hip joint with the abductor muscle weakness to a greater incidence of pain, arthritis, and lateral edge loading of the hip joint during repetitive walking up the ramps. Therefore, during walking patterns related to fast, normal, and slow gait, the highest pressure of moderate magnitude was positioned at the lateral roof of the acetabulum spanning the mid-stance phase.

The AQHF software can be useful in understanding the biomechanics of the hip joint contact forces. An insight into these forces is a necessary requirement for designing of the hip joint prosthesis. In a total hip replacement arthroplasty a part of the femur including the head of the femur is removed and an artificial ball is secured to the thigh bone [23]. The artificial socket must articulate with the ball component and should be fixed in a position to maintain an optimal JRF and GRF at the hip joint (Fig 9) [32].

## 4. Conclusion

In sum, the findings of our study show a significant difference (p < 0.05) between males and females in the magnitude of the hip JRF. Males exhibit a higher mean value of the hip JRF than

females during the single-limb support phase that represents the middle of the stance phase of the gait cycle varied from thirty up to sixty percentage at a ramps of 0°, 5°, 10°, and 15°. In the light of these findings, achieving a sufficient interface fit between the implant and the hip bone (Fig 10A and 10B) has important implications for prosthetics and assistive devices.

An increase in the mean value of the male hip JRF during the single-limb support phase while walking up the four tested ramps compared to the female value predicts a higher level of micromotion. This predicts for a greater femoral fracture of the male prosthetic hip compared to females [33,34]. Thus, calculations using the AQHF software can provide clinical values for the orthopaedic surgeons in predicting hip joint degenerating mechanism and prosthetic implant wear. Our software can calculate the active forces on normal individuals' hip joints and may help predict the possibility of wear and tear if the JRF has deviated from the normal value. Also, the hip replacement therapy and the use of the best implant material can be devised based on the results generated by our software to establish the best fit customized for each individual.

We concluded that an increase in the angle of the walking ramp is associated with an increase in the hip JRF. Consequently, it should be considered that males with a prosthetic hip joint may result in a higher incidence of femoral fracture of their prosthetic hip while walking up the ramps. The higher risk is attributed to an increase in the reaction force of the hip joint while walking up ramps of various inclinations. The software AQHF supports the evaluation of the JRF and GRF that can be used for diagnosis in clinical settings.

The advantage of this software is to predict the hip JRF in addition to GRF at the hip joint through three dimensional analysis in a critical sub-phase of the gait cycle i.e., the single limb support phase. This phase represents the least stable position of all phases of the gait cycle while walking and a critical point in gait cycle to exert maximum hip forces. This makes the AQHF software unique. Besides, there are no comparable softwares for calculating hip joint GRF and JRF in normal people that can be extrapolated to predict disease disposition. Hence, no similar softwares are reported that can precisely calculate these forces in gait phases while walking on level ground and during walking up ramps. The following model (Fig 12) determine how this software works depending on the input and output data.

## 5. Material and methods

All the participants included in this study were healthy, normal subjects with no disease or musculoskeletal deformities. A verbal consent was obtained from each participant. The participation in this study was based on voluntary enrollment. The study is solely a physical assessment study and no invasive procedure was done. The current research work was approved by Research Ethics Committee of Faculty of Physical Therapy of Cairo University. NO:P.T.REC/ 012/006572.

**Software Analysis Procedure:** Before starting a new analysis with the AQHF software, the following data input was required:

1. Subject data that include Weight, height, Age, Height, gender.)

2. The data files from the mathematical calculation included JRF and GRF data files.

3. The measurement report for the specified inclined surfaces.

4. The AQHF software requires input of the following steps for carrying out an analysis:

## 1. The start menu of the AQHF Application:

From this menu, all functions in the software can be reached.

Starting a new analysis: The File>detail option is selected (Fig 6A).

The Person Data menu is displayed. The relevant subject data is entered, and the files used as input for the analysis are selected (Fig 6B).

If the measurement report is filled in correctly, all relevant information is stored in the report. All data entered in the person data menu will be saved in the hip analysis file.

## 2. Type of Analysis:

This field determines whether the software will calculate either JRF or GRF.

## 3. Calculation and Results:

After the software is properly set up, the analysis is started from the Analysis menu. This menu has several options; JRF calculation and GRF calculation for 0,5,10 15 degrees (Fig 6C) and another option for new ramp estimation that can calculate up to 70 degrees of ramp inclination for both JRF and GRF (Fig 6D). When the calculations are completed, the results window shows the calculated hip force parameters.

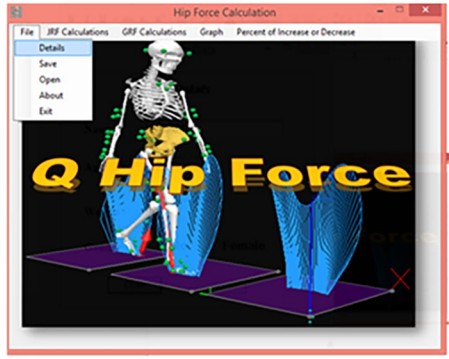
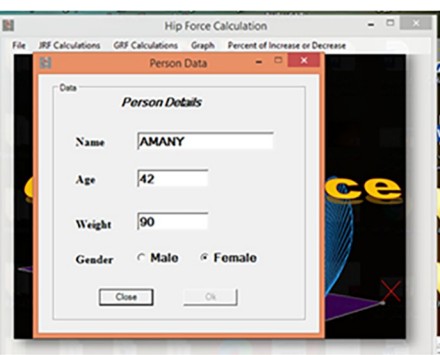

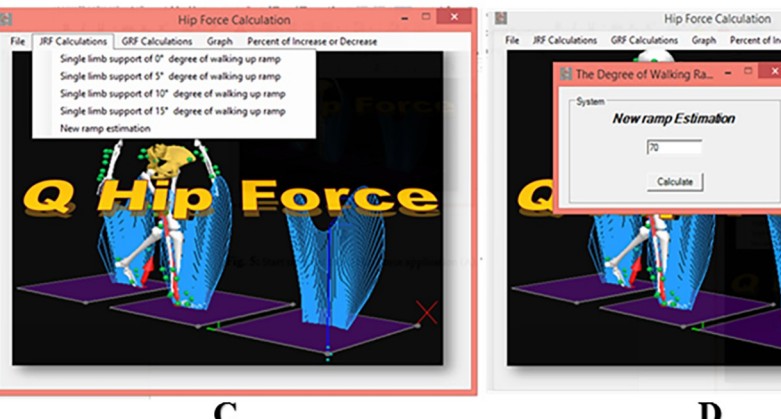

**Fig 6. Hip force calculation through AQHF. (A)** The start menu of the AQHF Application. The hip force calculation can be done through the Q hip force start window. All functions in the software can be reached through the start menu. The File>detail option is selected. (B) This displays the Person Data menu. The relevant subject data is entered, and the files used as input for the analysis are selected. (C) Calculation for 0,5,10 and15 degrees ramps is done through this window. (D) New ramp estimation that can calculate up to elevation of 70 degrees.

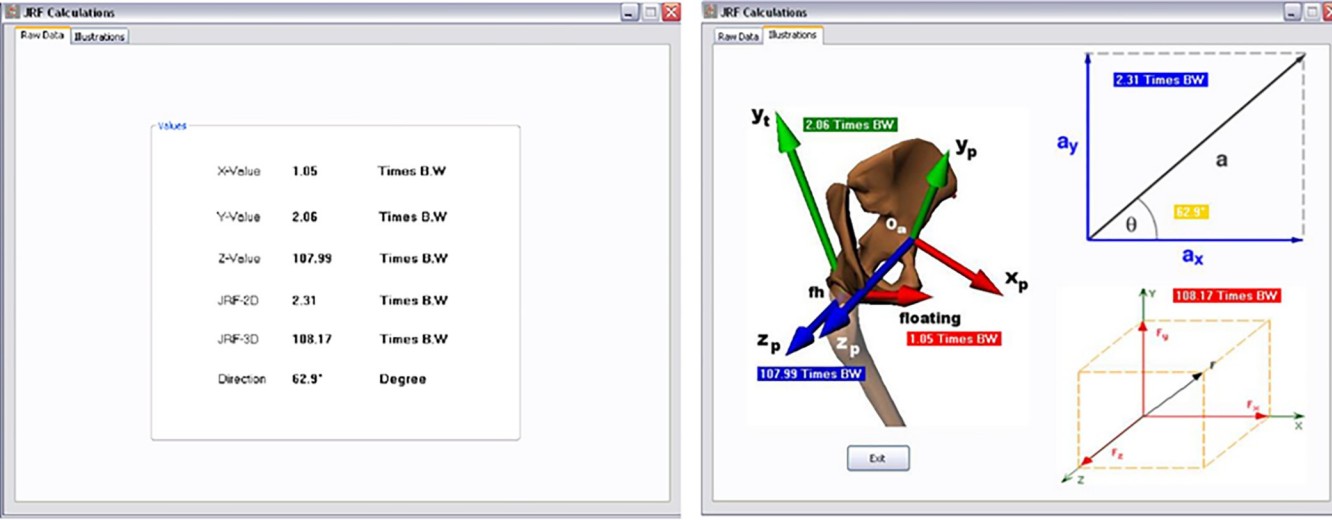

**A**                                                                                 **B**

**Fig 7. JRF calculation.** (A) The first window indicates Raw Data. (B) The second window is an illustration of the raw data indicated in window A.

By selecting the JRF calculation, a standard JRF for each inclined surface is displayed. There are two windows; the first one (Raw Data Fig 7A) displays the value of the hip JRF in each dimension, the 2D and 3D resultant of the hip JRF, and its direction. The second window illustrates the figures that show the values of the hip JRF (Fig 7B).

When GRF calculation is selected, two windows appear; the first one (Raw Data) (Fig 8A) displays the value of the GRF at the hip joint in each dimension, the 2D and 3D resultant of the GRF at the hip joint in addition to its direction. The second one displays the figures that show the values of the GRF at the hip joint in each direction (Fig 8B).

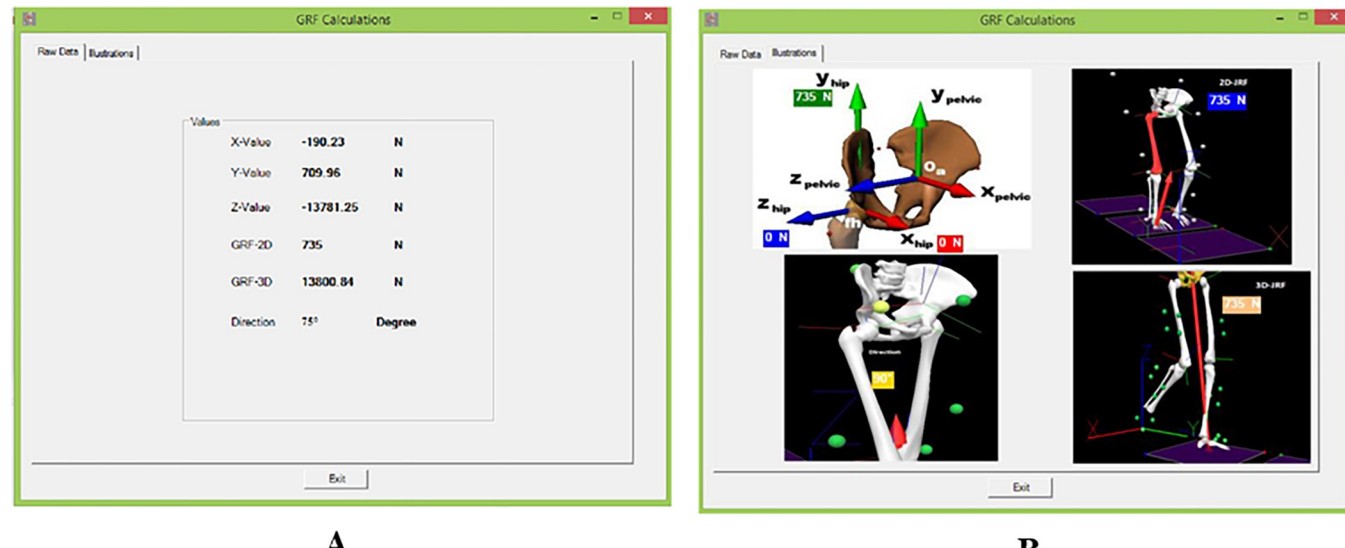

**A**                                                                                 **B**

**Fig 8. GRF calculation through AQHF.** (A) The first window indicates raw data. (B) The second window is an illustration of raw data shown in window A.

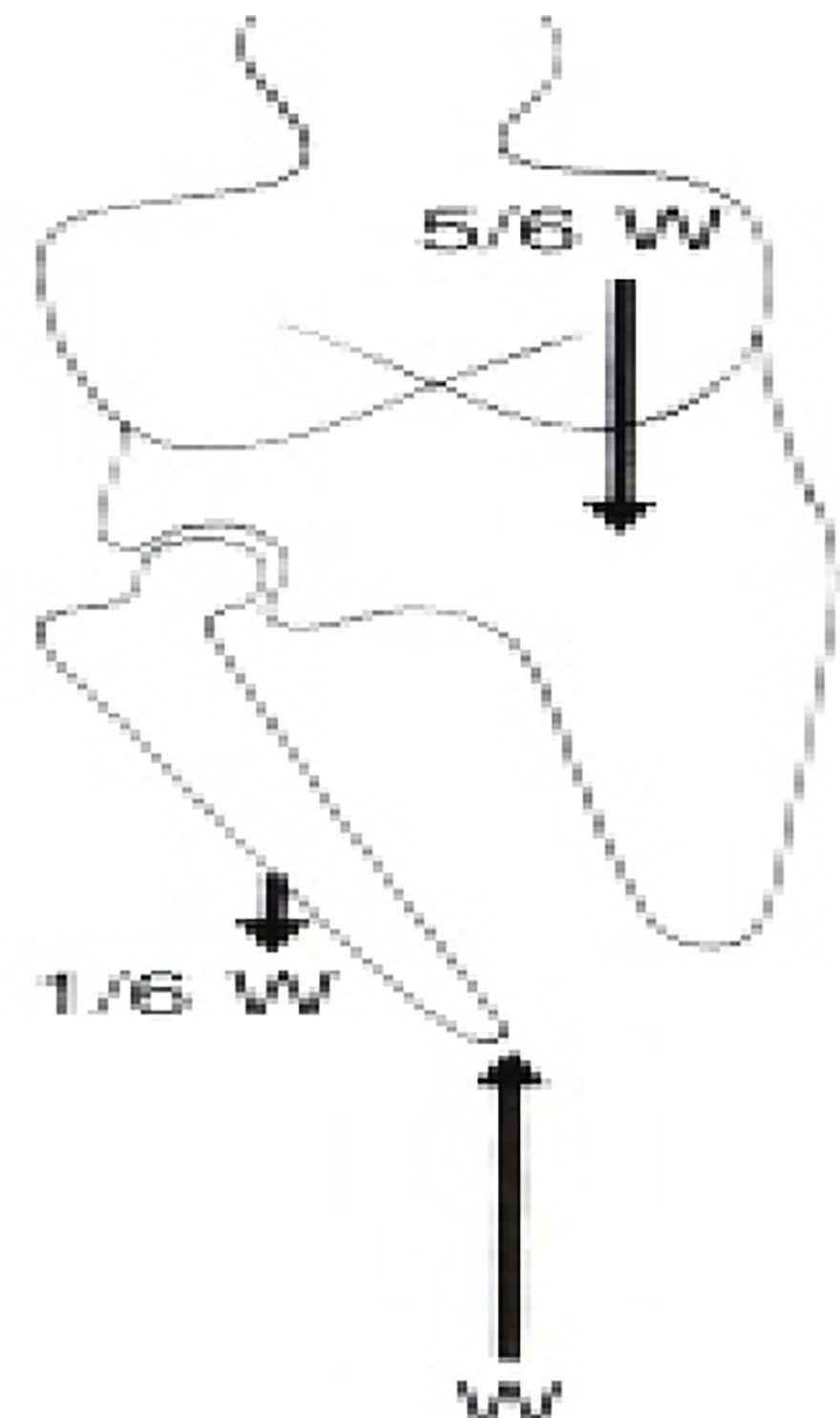

**Fig 9. Free body diagram showing the perpendicular orientation of GRF for the forces acting on the body.**
(Adapted from Frankel and Nordin 2001).

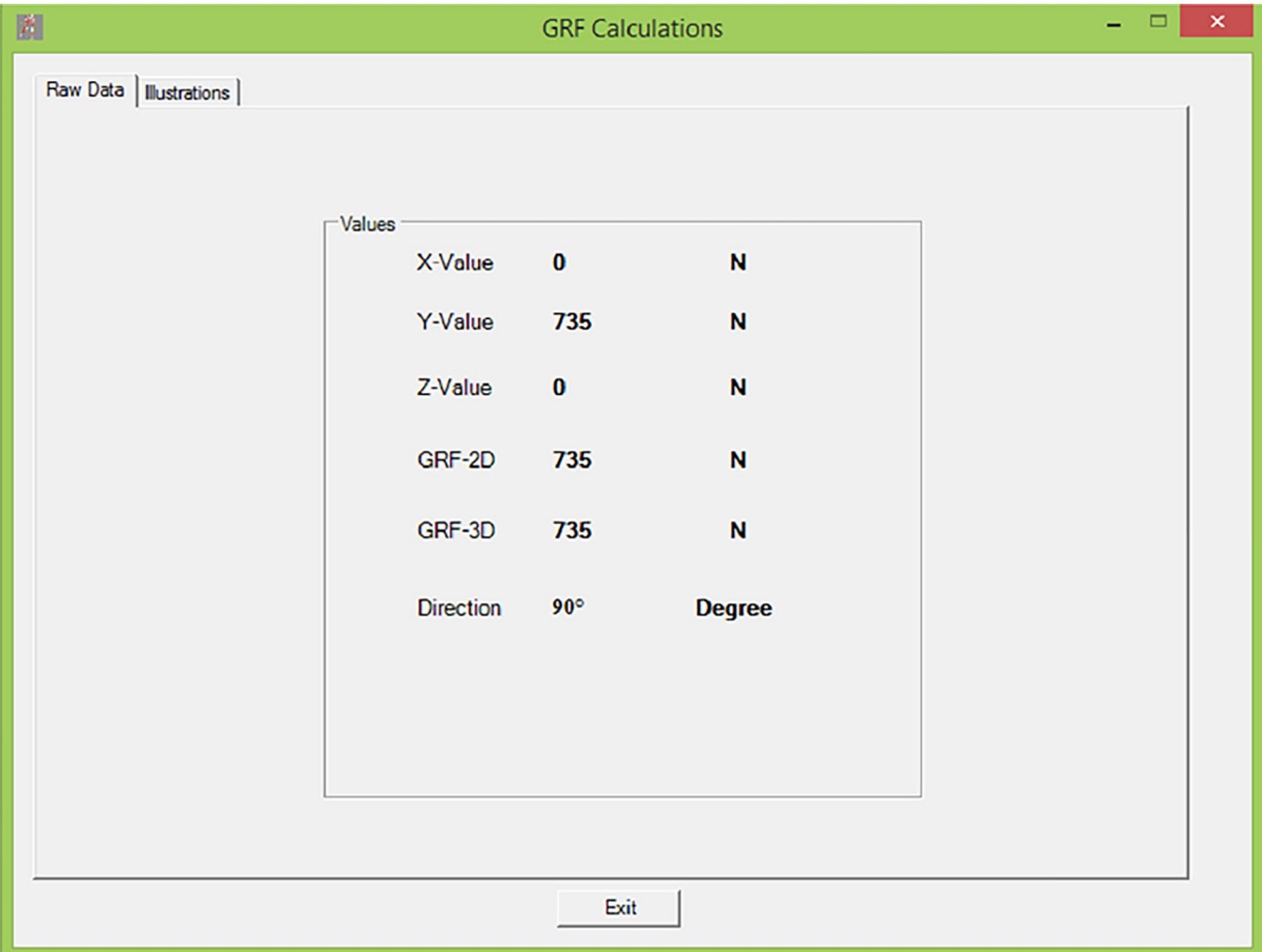

**Fig 10. GRF calculation.** The window demonstrates the raw data during the level surface walking. X-value and Z- value is zero.

As we previously mentioned that the direction of the GRF while walking on a level surface is 90° (referred to Fig 9), and the orientation of the GRF vector with the ground while walking is perpendicular. Thus the force has only one component in the Y-direction. (Fig 10) Therefore the output of the software related to the value of the GRF while walking on a leveled surface differs from walking up other degrees of ramps.

Moreover, the percent of an increase and decrease option has two functions; the first one calculates the percent of an increase or decrease in the mean value of the hip JRF or the GRF during the early part of the single-limb support phase on inclined surfaces. The second function calculates this ratio for each 2D and 3D resultant of the hip JRF and GRF. These two functions enable the users to calculate this percent when the ramp has a fixed degree and when the weight has a fixed value (Fig 11A and 11B). Thus the model presented in Fig 12 and the flowchart (Fig 13) demonstrate the synchronization between Q hip force software, Q gait and Q view softwares needed for calculating the GRF and JRF values, which are important for prediction and extrapolation of data. The result is derived from the input data through the computer programming languages of the software. This language uses different source codes to facilitate the formatting of these equations to be understood and manipulated in order to designing this new software.

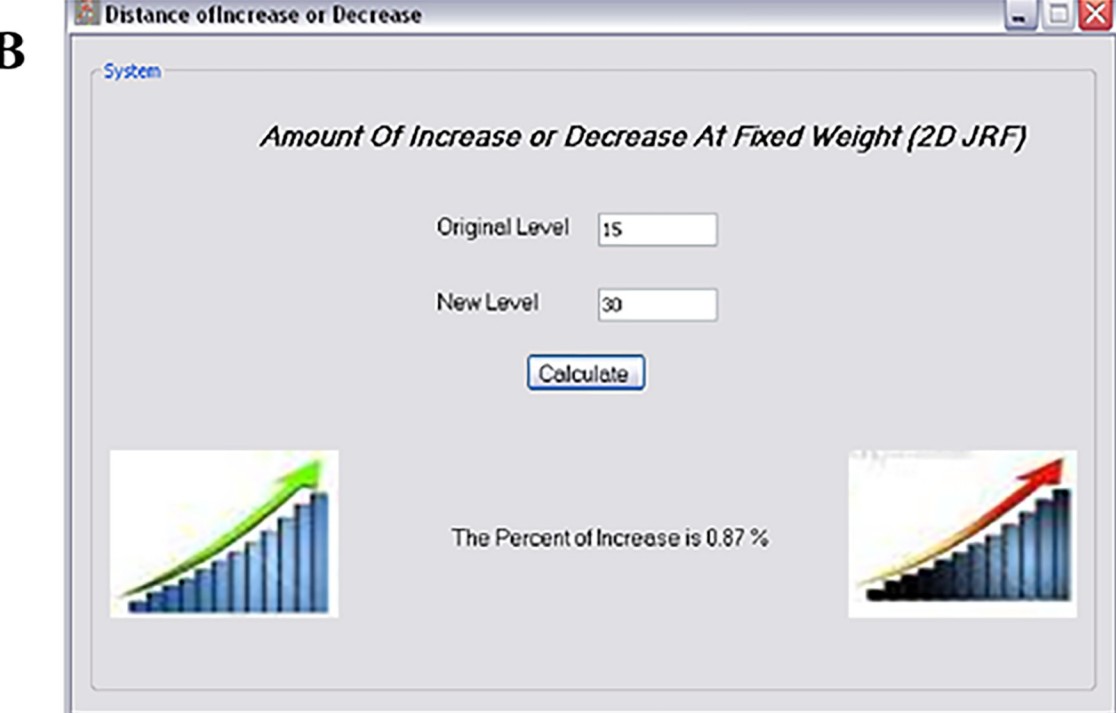

**Fig 11. The windows show the final results reflecting the increase or decrease in the GRF and JRF calculations.** (A) The percentage of increase or decrease in GRF at fixed ramp can be observed in this window and (B) the amount of increase and decrease in the 2D JRF at fixed weight can be observed at the window.

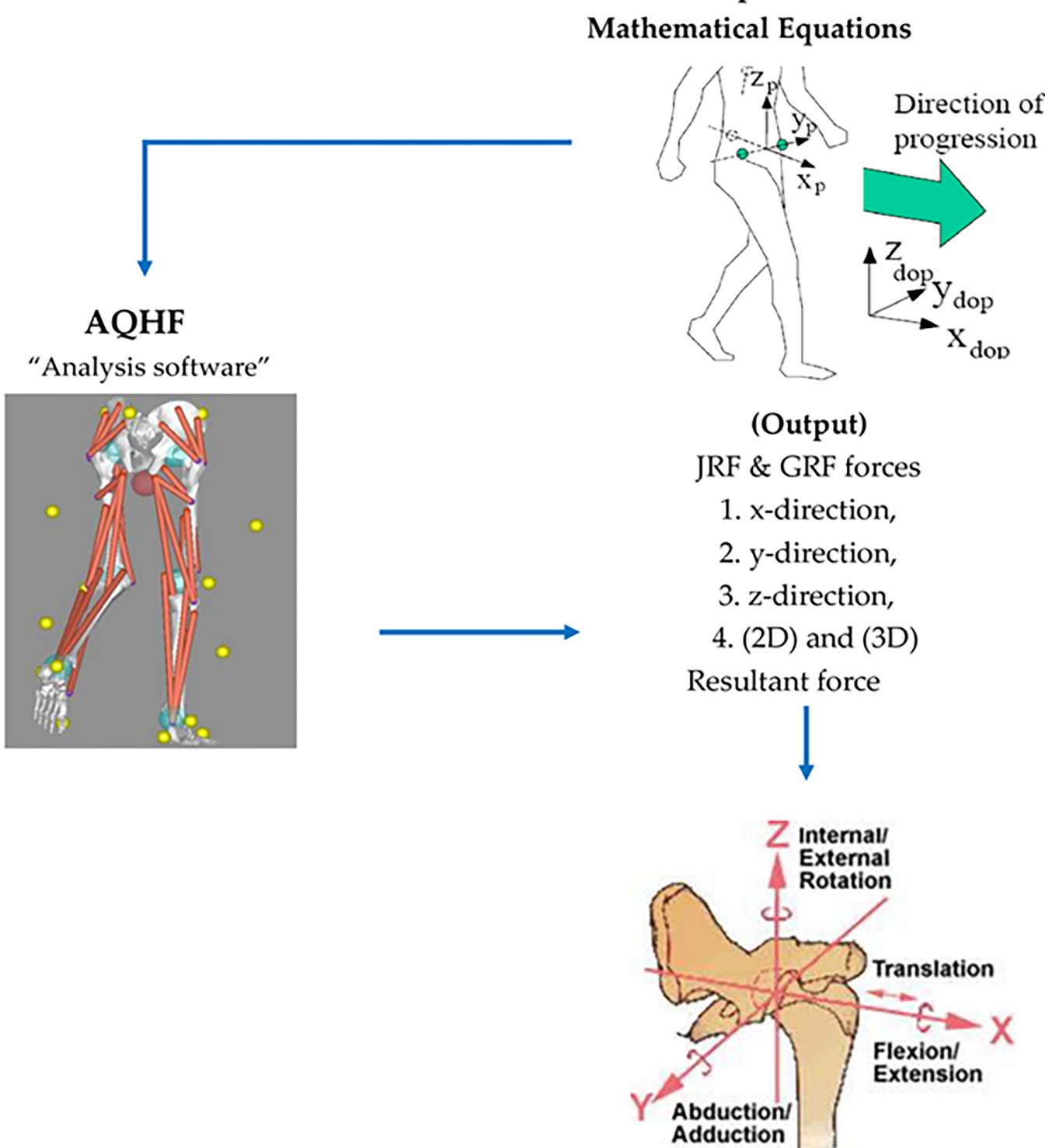

**Fig 12. The model shows the analytics of data input, calculation through AQHF software and the generation of results.** This mechanism is followed for calculation at each inclination.

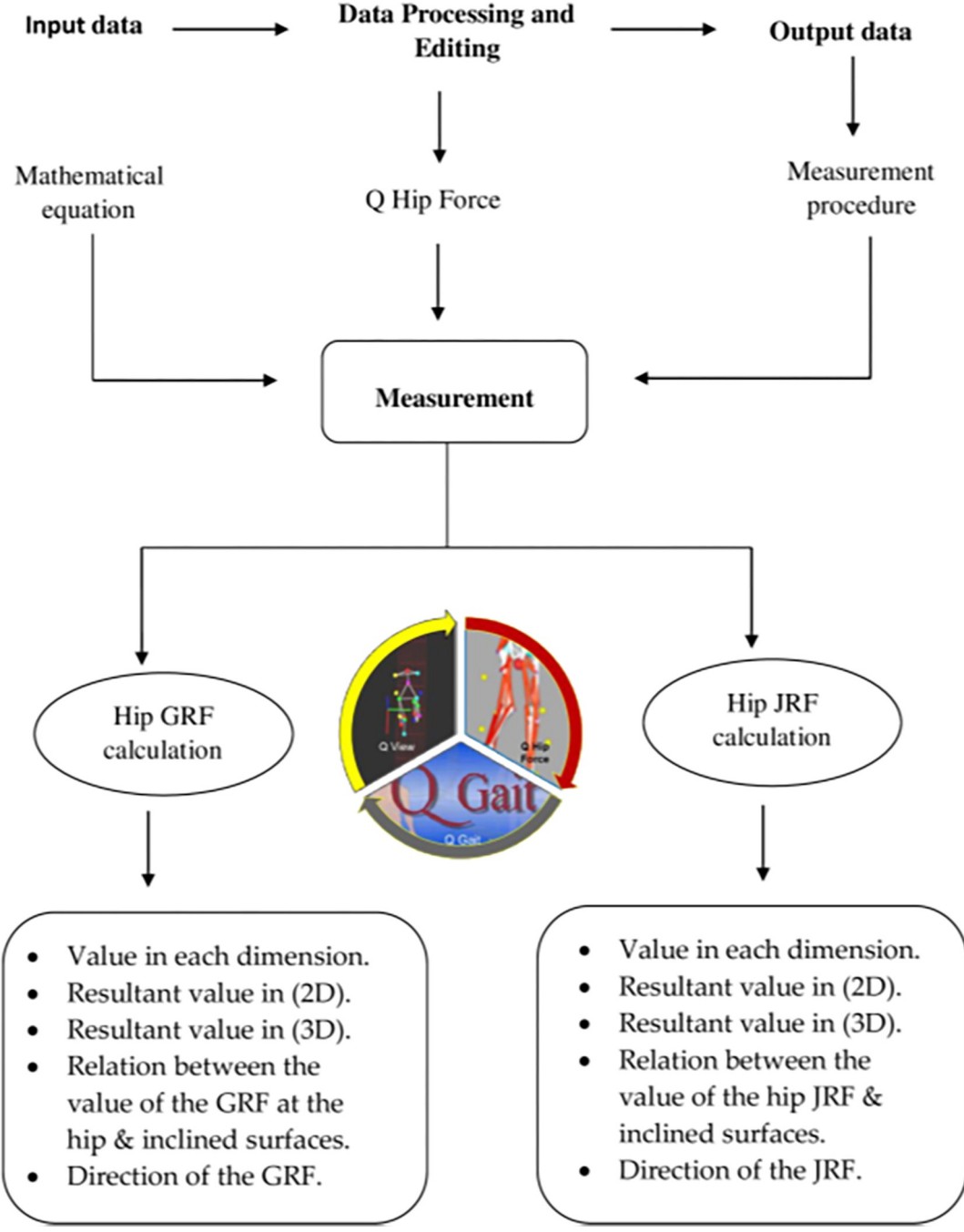

**Fig 13. The flowchart representing the step by step measurement of the GRF and JRF with the Q-gait and AHQF softwares, sequentially following the input phase, measurement phase and the output phase.**

The variables are derived from the data input of healthy subjects without any deformities and health issues, hence, the study may benefit somewhat from increasing the number of subjects from both gender. The strength of this study is that the software can effectively and precisely calculate upslope walking upto 70˚ without a person to actually go through upslope gaits to achieve an input value.

## Author Contributions

**Conceptualization:** Amany Eid Abd El-Tawab.

**Data curation:** Amany Eid Abd El-Tawab, Aisha Farhana.

**Formal analysis:** Aisha Farhana.

**Investigation:** Amany Eid Abd El-Tawab.

**Methodology:** Aisha Farhana.

**Software:** Amany Eid Abd El-Tawab.

**Validation:** Aisha Farhana.

**Writing – original draft:** Amany Eid Abd El-Tawab, Aisha Farhana.

**Writing – review & editing:** Amany Eid Abd El-Tawab, Aisha Farhana.

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
