## [Decision Letter · Decision Letter 0]

13 Jun 2022

PONE-D-22-13269Three Dimensional Analysis of Hip Joint Reaction Force using Q Hip Force (AQHF) Software: Implication as a Diagnostic ToolPLOS ONE

Dear Dr. Farhana,

Thank you for submitting your manuscript to PLOS ONE. After careful consideration, we feel that it has merit but does not fully meet PLOS ONE’s publication criteria as it currently stands. Therefore, we invite you to submit a revised version of the manuscript that addresses the points raised during the review process.

We look forward to receiving your revised manuscript.

Kind regards,

Ghulam Md Ashraf, Ph.D.

Academic Editor

PLOS ONE

Journal Requirements:

2. Please ensure you have stated in your manuscript text the full formal name of the ethics committee that approved this study.

Please also provide additional details regarding participant consent. In the ethics statement in the Methods and online submission information, please ensure that you have specified (1) whether consent was informed and (2) what type you obtained (for instance, written or verbal, and if verbal, how it was documented and witnessed). If your study included minors, state whether you obtained consent from parents or guardians. If the need for consent was waived by the ethics committee, please include this information.

Additionally, please note that PLOS ONE has specific guidelines on code sharing for submissions in which author-generated code underpins the findings in the manuscript. In these cases, all author-generated code must be made available without restrictions upon publication of the work. Please review our guidelines at https://journals.plos.org/plosone/s/materials-and-software-sharing#loc-sharing-code and ensure that your code is shared in a way that follows best practice and facilitates reproducibility and reuse.

5. Please include a caption for figure 11.

Reviewers' comments:

Reviewer's Responses to Questions

**Comments to the Author**

1. Is the manuscript technically sound, and do the data support the conclusions?

Reviewer #1: Yes

Reviewer #2: Yes

Reviewer #3: Yes

Reviewer #4: Yes

2. Has the statistical analysis been performed appropriately and rigorously? 

Reviewer #1: Yes

Reviewer #2: Yes

Reviewer #3: Yes

Reviewer #4: Yes

3. Have the authors made all data underlying the findings in their manuscript fully available?

Reviewer #1: Yes

Reviewer #2: Yes

Reviewer #3: Yes

Reviewer #4: Yes

4. Is the manuscript presented in an intelligible fashion and written in standard English?

Reviewer #1: Yes

Reviewer #2: Yes

Reviewer #3: Yes

Reviewer #4: Yes

5. Review Comments to the Author

Reviewer #1: Inclusion criteria of Participants should be added.

Assigned Consent Form by each participant should be mentioned in the manuscript.

Ethical Committe number and site should be added to the manuscript.

Limitation section of the study should be added.

Reviewer #2: In this study, the authors designed a software, the Analysis Q Hip Force (AQHF), to analyze the data retrieved from the mathematical equations for calculating the JRF and ground reaction force (GRF) that act on the hip joint during the early part of the stance phase. The paper is very interesting and technically sound.

1) The references is up to date

2) The manuscript is grammatical correct

3) The figures, tables and the statistics are satisfactory.

Major Comments:

1) The authors must write a small section describing the technical characteristics of the AQHF software and the complexity of the functions

2) The authors must analyze advantages and disadvantages compared to other similar software

Reviewer #3: The article is quite interesting trying to calculate the active forces on hip joints that might help in predicting the wear and tear for deviated JRF values. This might help in design of the implant material for a customized fit. With personalized medical devices being the future, this article is an important step towards that. The article is clinically relevant to bring a change and add dimensions to the current therapies for hip pain and gait-associated afflictions. It may be helpful when considering clinical decision-making.

However, I would like the authors to address a few points:

1. The authors mention that the software can predict a healthy gait index with input readings up to 70� elevations. This requires some explanation as to how the software can calculate the extrapolations with precision. Hence, in Section 5, the third point in the material and method section requires a descriptive explanation as to how the Q hip force software synchronizes with the AQHF software to predict the results with significance. An illustration showing this synchronization can be added.

2. In Sections 3.2 and 4, the authors have mentioned an increase in the mean value of male hip JRF compared to the females during walking up four tested elevations. They conclude that this indicates a high degree of micro-motion. Though these results are calculated through software and statistical methods, the percentage of the gait cycle should be mentioned when calculating both ground reaction force and joint reaction force, to affirmatively conclude the results. Hence, the percentage of GRF and JRF must be added in section 3.2 and 4, and should be included in the downstream calculations of the results.

3. In section 2, the authors have discussed the results of two parameters, section 2.1 discusses the effect of gender (Factor B) on 3D hip JRF obtained from AQHF Software, and section 2.2 demonstrates the interaction effect of ramps and gender on the resultant hip JRF. However, no values were provided for the interaction effect of ramp and gender on the resultant GRF. Since both GRF and JRF values are important for prediction and extrapolation of data till 70� elevations, it is mandatory to explain the interaction effect of ramp and gender on GRF as well.

Minor points:

a. Line numbers must be added for ease of reading and tracking.

b. Typographical errors must be corrected.

c. Use uniform format for writing “Tables and Figures”. In some places it is mentioned as fig or figure, and in others as Fig.

Reviewer #4: The strength of the study is the analysis of both JRF and GRF reaction forces for females and males, taking into account the structural differences in both genders. Furthermore, the study is based on the data calculated by actual analysis of the subjects which is used as input data in the Q hip software, with subsequent calculations done through AQHF software. However, some points need to be addressed.

1. The study has used upslope walking instead of level-ground walking. Since the usual walking pattern mostly involves level-ground walking, why is leveled walking not used in the study instead upslope walking is used to calculate the hip joint force. The authors should explain why they have not used ground-level walking.

2. In the introduction section, the authors mention that they have used the motion analysis system which utilizes marker-based motion planes and a force platform to record the kinematic and kinetic gait parameters in a subject-specific investigation method. However, they have not mentioned the frequency at which the data was recorded. Since the analysis depends on the frequency of each data point to obtain kinetic gait parameters, the study requires a clear indication of frequencies for data point calculations.

3. It is evident that the GRF and JRF calculations are used as a final input in the AQHF software, which calculates the actual hip-joint forces. In the results section, the authors have calculated the effect of gender on the hip JRF (section 2.1) and the effect of ramps and gender on the resultant hip JRF (section 2.2). There is no mention of the effect of ramp and gender on the final GRF. This seems to be missing in the study and must be incorporated and discussed.

4. Figure 7 is unclear and should be replaced by a clearer picture.

Minor points:

1. Line numbers can be added for easier readability.

2. Relevant latest references from 2022 can be added (e.g doi: 10.1016/j.gaitpost.2014.06.013).

3. Though some studies have developed prediction models to calculate spinal and joint forces in certain disabilities and amputations, the present study evaluates the forces in normal subjects to identify a healthy walking index. Any derangement from normal forces can be easily identified by the results calculated through the AQHF software. Hence, this study fills a lacuna in synchronizing the research results and its feasible use in clinical settings.

4. The mathematical calculation backed by descriptive statistics with LSD and factorial design provides a rigorous analysis of the subject-based results to be used as input for the software.

5. The study is promising to provide sufficient grounds for further studies that can facilitate the routine use of the AQHF software in clinical settings.

6. PLOS authors have the option to publish the peer review history of their article (what does this mean?). If published, this will include your full peer review and any attached files.

Reviewer #1: **Yes: **Asmaa M. Elbandrawy

Reviewer #2: No

Reviewer #3: **Yes: **M. Dharma Prasad

Reviewer #4: No

---

## [Author Response · Author response to Decision Letter 0]

27 Jul 2022

We thank the reviewers for their constructive comments that improved the quality of the paper. The entire manuscript has been revised accordingly and the point to point responses to the comments are as follows. The significant revisions are highlighted.

Reviewer 1:

Point-by-point reply 

Comment: Inclusion criteria of Participants should be added.

Response: The inclusion and exclusion criteria are added in section 5 (material and methods); lines 404-405.

Comment: Assigned Consent Form by each participant should be mentioned in the manuscript.

Response: The study includes healthy subject and is solely a physical assessment study with no invasive procedure involved. A verbal consent was obtained from each participant and the participation was based on voluntary enrollment. This paragraph has been included in section 5 (material and methods); lines 404-409.

Comment: Ethical Committee number and site should be added to the manuscript.

Response: The aforementioned has been added in section 5 (material and methods); lines 407-409.

Comment: Limitation section of the study should be added.

Response: Thank you very much for your comment. It helped to clearly write about the strength and limitations of the study. 

Since the input variables of the study are based on healthy subjects, increasing a sample size is likely to have little benefit to the overall calculation methods and results. Software-wise our application has been tested thoroughly to comply with the language variables and stringency. Hence, it is highly unlikely that there is a limitation in this study on both fronts ie., sample size, methodology or programming. 

This description is included in the section 5, material and methods (line 449-455).

Reviewer 2:

Point-by-point reply 

Comment: In this study, the authors designed a software, the Analysis Q Hip Force (AQHF), to analyze the data retrieved from the mathematical equations for calculating the JRF and ground reaction force (GRF) that act on the hip joint during the early part of the stance phase. The paper is very interesting and technically sound.

 1) The references is up to date

 2) The manuscript is grammatical correct

 3) The figures, tables and the statistics are satisfactory.

 Response: We thank the reviewer for the positive comment. 

Comment: The authors must write a small section describing the technical characteristics of the AQHF software and the complexity of the functions.

Response: Thank you for the constructive comment. The information has been added in section 5 (material and methods) describing the programming method and input variables (line 456-460). Additionally we have added illustration in the form of model (Fig. 12) and flowchart (Fig.13) to demonstrate the synchronization between input data – software based data measurement and the generation of results as output data.

Comment: The authors must analyze advantages and disadvantages compared to other similar software.

Response: Thank you for your comment. We have added a paragraph in section 4 (conclusions) line 392-400.

Reviewer 3:

Point-by-point reply 

Comment: The authors mention that the software can predict a healthy gait index with input readings up to 70� elevations. This requires some explanation as to how the software can calculate the extrapolations with precision. Hence, in Section 5, the third point in the material and method section requires a descriptive explanation as to how the Q hip force software synchronizes with the AQHF software to predict the results with significance. An illustration showing this synchronization can be added. GRF and JRF values are important for prediction and extrapolation of data till 70� elevations.

Response; Thank you very much for your comment to add clarity to the methodology. We have now incorporated the detail in section 5 (line 449-455), and also added a model (Fig. 12) and flowchart (Fig.13) to describes the sequence of program. 

Comment: In Sections 3.4 and 4, the authors have mentioned an increase in the mean value of male hip JRF compared to the females during walking up four tested elevations. They conclude that this indicates a high degree of micro-motion. Though these results are calculated through software and statistical methods, the percentage of the gait cycle should be mentioned when calculating both ground reaction force and joint reaction force, to affirmatively conclude the results. Hence, the percentage of GRF and JRF must be added in section 3.2 and 4, and should be included in the downstream calculations of the results.

Response: The percentage is included now in both the section. In section 3.4 the sentence (line 310-314) have been rewritten to incorporate the percentages, and in section 4 a sentence has been added (Line 360-361).

Comment: In section 2, the authors have discussed the results of two parameters, section 2.1 discusses the effect of gender (Factor B) on 3D hip JRF obtained from AQHF Software, and section 2.2 demonstrates the interaction effect of ramps and gender on the resultant hip JRF. However, no values were provided for the interaction effect of ramp and gender on the resultant GRF. Since both GRF and JRF values are important for prediction and extrapolation of data till 70� elevations, it is mandatory to explain the interaction effect of ramp and gender on GRF as well.

Response: The reviewer rightly pointed out that GRF is affected by the modulation in the ramp elevation and gender hence, it is discussed extensively in the discussion section 3.4. 

Minor Comments: 

a. Line numbers must be added for ease of reading and tracking.

b. Typographical errors must be corrected.

c. Use uniform format for writing “Tables and Figures”. In some places it is mentioned as fig or figure, and in others as Fig.

Response: The above points have been addressed. 

Reviewer 4:

Point-by-point reply 

Comment: The strength of the study is the analysis of both JRF and GRF reaction forces for females and males, taking into account the structural differences in both genders. Furthermore, the study is based on the data calculated by actual analysis of the subjects which is used as input data in the Q hip software, with subsequent calculations done through AQHF software. 

Response: We are thankful for the above comment. 

Comment: The study has used upslope walking instead of level-ground walking. Since the usual walking pattern mostly involves level-ground walking, why is leveled walking not used in the study instead upslope walking is used to calculate the hip joint force. The authors should explain why they have not used ground-level walking. 

Response: The upslope walking induces more stress on hip joints as compared to levelled ground walking. In our study, we have included zero inclination, which is equivalent to levelled ground walking. Furthermore, elevations starting from 5� to 15� are upslope. Walking along the slopes induces more stress on the joint even in healthy individuals, hence this calculation allows the software to assess any underlying deviations from normal values. This adds a dimension for the effective use of this software in clinical settings

Comment: In the introduction section, the authors mention that they have used the motion analysis system which utilizes marker-based motion planes and a force platform to record the kinematic and kinetic gait parameters in a subject-specific investigation method. However, they have not mentioned the frequency at which the data was recorded. Since the analysis depends on the frequency of each data point to obtain kinetic gait parameters, the study requires a clear indication of frequencies for data point calculations.

Response: A description is now added to the Introduction Section (section -1)- line and 70-73. 

Comment: It is evident that the GRF and JRF calculations are used as a final input in the AQHF software, which calculates the actual hip-joint forces. In the results section, the authors have calculated the effect of gender on the hip JRF (section 2.1) and the effect of ramps and gender on the resultant hip JRF (section 2.2). There is no mention of the effect of ramp and gender on the final GRF. This seems to be missing in the study and must be incorporated and discussed.

Response: The reviewer rightly pointed out that GRF is affected by the modulation in the ramp elevation and gender. It is discussed extensively in the discussion section 3.4. 

Comment: Figure 7 is unclear and should be replaced by a clearer picture.

Response: The picture has been replaced now. 

Minor points:

Comment: Minor points:

1. Line numbers can be added for easier readability.

2. Relevant latest references from 2022 can be added (e.g doi: 10.1016/j.gaitpost.2014.06.013)

Response: The line numbers and the latest reference is now added. 

Comment: Though some studies have developed prediction models to calculate spinal and joint forces in certain disabilities and amputations, the present study evaluates the forces in normal subjects to identify a healthy walking index. Any derangement from normal forces can be easily identified by the results calculated through the AQHF software. Hence, this study fills a lacuna in synchronizing the research results and its feasible use in clinical settings.

Response: We thank the reviewer for the comment.

Comment: The mathematical calculation backed by descriptive statistics with LSD and factorial design provides a rigorous analysis of the subject-based results to be used as input for the software.

Response: We thank the reviewer for the comment.

Comment: The study is promising to provide sufficient grounds for further studies that can facilitate the routine use of the AQHF software in clinical settings.

Response: We thank the reviewer for the comment.

---

## [Decision Letter · Decision Letter 1]

4 Aug 2022

Three Dimensional Analysis of Hip Joint Reaction Force using Q Hip Force (AQHF) Software: Implication as a Diagnostic Tool

PONE-D-22-13269R1

Dear Dr. Farhana,

We’re pleased to inform you that your manuscript has been judged scientifically suitable for publication and will be formally accepted for publication once it meets all outstanding technical requirements.

Kind regards,

Ghulam Md Ashraf, Ph.D.

Academic Editor

PLOS ONE

Additional Editor Comments (optional):

Reviewers' comments:

Reviewer's Responses to Questions

**Comments to the Author**

1. If the authors have adequately addressed your comments raised in a previous round of review and you feel that this manuscript is now acceptable for publication, you may indicate that here to bypass the “Comments to the Author” section, enter your conflict of interest statement in the “Confidential to Editor” section, and submit your "Accept" recommendation.

Reviewer #1: All comments have been addressed

Reviewer #2: All comments have been addressed

Reviewer #3: All comments have been addressed

2. Is the manuscript technically sound, and do the data support the conclusions?

Reviewer #1: Yes

Reviewer #2: Yes

Reviewer #3: Yes

3. Has the statistical analysis been performed appropriately and rigorously? 

Reviewer #1: Yes

Reviewer #2: Yes

Reviewer #3: Yes

4. Have the authors made all data underlying the findings in their manuscript fully available?

Reviewer #1: Yes

Reviewer #2: Yes

Reviewer #3: Yes

5. Is the manuscript presented in an intelligible fashion and written in standard English?

Reviewer #1: Yes

Reviewer #2: Yes

Reviewer #3: Yes

6. Review Comments to the Author

Reviewer #1: I thank the authors, all recommendations had been done .This is an interesting study and the authors have collected a unique dataset using cutting

edge methodology. The paper is generally well written and structured.

Reviewer #2: The authors have improved and revised the manuscript. It is technically sound and suitable for publication.

Reviewer #3: All questions and comments that I and other reviewers raised and communicated to authors are addressed to my satisfaction.

7. PLOS authors have the option to publish the peer review history of their article (what does this mean?). If published, this will include your full peer review and any attached files.

Reviewer #1: No

Reviewer #2: **Yes: **Athanasios Alexiou

Reviewer #3: No

---

## [Editor Report · Acceptance letter]

29 Aug 2022

PONE-D-22-13269R1 

Three Dimensional Analysis of Hip Joint Reaction Force using Q Hip Force (AQHF) Software: Implication as a Diagnostic Tool 

Dear Dr. Farhana:

I'm pleased to inform you that your manuscript has been deemed suitable for publication in PLOS ONE. Congratulations! Your manuscript is now with our production department. 

Kind regards, 

on behalf of

Dr. Ghulam Md Ashraf 

Academic Editor

PLOS ONE